# Fungal Diversity in Lichens: From Extremotolerance to Interactions with Algae

**DOI:** 10.3390/life8020015

**Published:** 2018-05-22

**Authors:** Lucia Muggia, Martin Grube

**Affiliations:** 1Department of Life Sciences, University of Trieste, via Licio Giorgieri 10, 34127 Trieste, Italy; 2Institute of Biology, Karl-Franzens University of Graz, Holteigasse 6, 8010 Graz, Austria; martin.grube@uni-graz.at

**Keywords:** cultures, metabarcoding, mycobiome, photobionts, phylogenetics, symbiosis, systematics

## Abstract

Lichen symbioses develop long-living thallus structures even in the harshest environments on Earth. These structures are also habitats for many other microscopic organisms, including other fungi, which vary in their specificity and interaction with the whole symbiotic system. This contribution reviews the recent progress regarding the understanding of the lichen-inhabiting fungi that are achieved by multiphasic approaches (culturing, microscopy, and sequencing). The lichen mycobiome comprises a more or less specific pool of species that can develop symptoms on their hosts, a generalist environmental pool, and a pool of transient species. Typically, the fungal classes Dothideomycetes, Eurotiomycetes, Leotiomycetes, Sordariomycetes, and Tremellomycetes predominate the associated fungal communities. While symptomatic lichenicolous fungi belong to lichen-forming lineages, many of the other fungi that are found have close relatives that are known from different ecological niches, including both plant and animal pathogens, and rock colonizers. A significant fraction of yet unnamed melanized (‘black’) fungi belong to the classes Chaethothyriomycetes and Dothideomycetes. These lineages tolerate the stressful conditions and harsh environments that affect their hosts, and therefore are interpreted as extremotolerant fungi. Some of these taxa can also form lichen-like associations with the algae of the lichen system when they are enforced to symbiosis by co-culturing assays.

## 1. Introduction

Lichens are long-living, self-sustaining, symbiotic systems that derive from mutualistic associations between biotrophic fungi (the mycobionts) and photosynthetic microorganisms (the photobionts, e.g., chlorophytes and/or cyanobacteria). Lichens represent one of the oldest known and recognizable examples of symbioses. They are commonly characterized by their dual nature, that is, composed of one mycobiont and a population of photobionts [1], which is wrapped by the fungal hyphae. However, this simplistic view of a dual symbiosis has been revised by recent studies, which revealed lichens as open houses for many other microorganisms, such as bacteria [2,3,4,5,6,7], additional algae [8,9] and fungi [10,11,12,13,14,15,16,17]. As a matter of fact, the fungal infections of lichens (Figure 1) were described even before lichens were recognized as a fungal-algal symbiosis (e.g., [18]). More than 1800 species of lichenicolous (meaning lichen-colonizing) fungi are known by scientific names. The majority of species are highly specific for their host, and until now they have mainly been classified by their reproductive characters and the symptoms that are caused on their host lichens [10,19]. On the other hand, certain features seem to facilitate the colonization by numerous lichenicolous fungi on the same host lichen. For example, 45 lichenicolous fungal species have been identified on the thalli of the common lichen *Xanthoria parietina* [20,21,22] and at least 11 species specifically infect *Tephromela atra* [23].

Only few lichenicolous fungi aggressively destroy their host and may cause the death of the thalli. These are usually unrelated to lichen-forming fungal lineages (and includes for instance aggressive members of Hypocreales or certain basidiomycetes). The majority of the lichenicolous fungi instead form local infections or live together with their hosts as commensals, and they are usually closely related to the lichen-forming fungal lineages [24,25,26]. Most of them seem to have evolved strategies to leave the fungal structure of the hosts unaffected, while taking the least benefit from algal photosynthates in order to support their mycelial structures. These and other microscopic studies also revealed other mycelial structures in lichens, which did not belong to infections of the recognized lichenicolous fungi. For a long time experts have therefore expected that lichens harbor a much larger number of so-called endolichenic fungi that cannot be estimated from externally visible infections.

Lichenicolous fungi and the mostly asymptomatic endolichenic fungi form the lichen mycobiome. The lichen mycobiome comprise stable and transient guilds, which, to some extent, correlate with the ecological conditions of the lichen habitats. Lichens from humid, temperate, and boreal environments mainly host fungi representing the classes Sordariomycetes and Leotiomycetes; these lineages are close relatives of plant endophytes [11,13]. Rock-inhabiting lichens, which are often exposed to fluctuations of temperature and humidity, are rather colonized by melanized fungi [27,28]. These fungi, comprising unknown and known hyphomycetous lichenicolous fungi, show close affinities to non-lichenized, extremotolerant rock-inhabiting fungi, from oligotrophic environments, and to plant and animal pathogenic black yeasts in the classes Dothideomycetes and Eurotiomycetes [15]. They are widely known as black fungi because they accumulate melanins in their cell walls. Melanins usually confer them the ability to grow in oligotrophic environments and the resistance to multiple abiotic stresses (such as high doses of radiation, desiccation, and temperature extremes) [29]. Black fungi are, therefore, usually recognized as (poly)extremotolerant organisms. Under these conditions, black fungi also develop plastic growth strategies (e.g., switching between filamentous, microcolonial, or yeast growth) and usually do not produce energy-demanding, sexually reproductive structures or rely on asexual reproduction [30]. They do not necessary only develop their mycelia on open surfaces but they are able to stretch their hyphae into fine rock crevices and are also able to penetrate the substrate down to few millimetres below the surface, to form endolithic communities [31]. In these environments, black fungi can co-occur with other stress-tolerant microorganisms, such as cyanobacteria or aerial green algae [32,33].

Here, we review the knowledge about these lichen-associated fungi, as well as the traditional and most recent approaches that are used to study the diversity of lichen mycobiomes, including phylogenetics, community sequencing, microscopy, chemical analyses, and culture experiments. We also draw attention to some taxa, which have the capacity to form in vitro lichen-like associations with the algae of the lichen system when they are enforced to symbiosis by co-culture.

## 2. Diversity of the Lichen Mycobiome as Revealed by High-Throughput Sequencing

Initially, the culture-independent community fingerprint analyses helped to uncover the dimension of the lichen-associated mycobiome [12,14]. In the past few years, high throughput sequencing has become a standard for the description of microbial diversity and functionality. Lichens are, for their dominant part, developed and shaped from the structures of their lichenized fungal partner (mycobiont). The mycobiont DNA also represents the most abundant fraction of reads in the fungal high throughput sequencing approaches (e.g., [6,34,35,36]). However, largely varying abundances in the samples (as well as the nature of sequencing technologies) can lead to biased estimates of community richness and composition [37]. To avoid the loss of information regarding the fungi at low abundances, special care has to be taken in screening and rarefying the sequence data before the statistical analyses. Since the relative quantity of the lichen mycobiont reads are unpredictably high, U’Ren et al. [38] has used species-specific blocking primers to prevent the amplification of mycobiont DNA in the analyses of the residual mycobiome. For the efforts that are associated with the application of mycobiont-specific blocking primers, this approach might be less suitable for the analyses of larger numbers of host lichen species, but it is highly recommended for larger sample sizes corresponding to the same lichen mycobiont.

Metabarcoding studies (fungal community analyses) of lichen mycobiomes have previously relied on the 454 pyrosequencing platform and used specific primers to target the nuclear internal transcribe spacer region (ITS) [13,34,35]. The ITS locus has been proposed as the best performing fungal barcode [39] and ITS sequence data have proved suitable to delimit the molecular operational taxonomic units (MOTUs) in most studies, including the lichenized and non-lichenized fungi [13,34,35,40,41]. However, because of technical restrictions on the sequence lengths, either the ITS1 or ITS2 subregions were sequenced. The analysis of the individually sequenced fragments of the same material shows that the estimates of diversity are not fully congruent [41,42], mainly because of reasons such as the higher variability of ITS2, the still insufficient surface desinfection procedures in the metabarcoding approaches, and the known shortcomings of clustering analyses. In addition, the details of storage of the lichen material prior to the DNA extraction, amplification, as well as sequence quality control can affect the estimation of fungal diversity [38].

In their study on the mycobiome in mosses and lichens, U’Ren et al. [13] recovered only members of Pezizomycotina, with the most represented classes being Sordariomycetes and Leotiomycetes. However, there was also an ecologically flexible group of symbionts that occurred both as the endolichenic fungi and as endophytes of mosses. Muggia et al. [15] observed similar patterns by detecting strains that belonged to Epibryaceae (Chaetothyriales, Eurothiomycetes) from the alpine, epilithic lichens. Interestingly, fungi of this family includes numerous endophytic taxa, which are symptomatic, and form ascomata on their bryophyte hosts [43,44]. Beside these, only a few studies are so far available, which have used next generation sequencing methods to describe the fungal diversity in lichens. Zhang et al. [34] assessed the diversity and distribution of the fungal communities that were associated with seven lichens in the Ny-Ålesund Region (Svalbard, High Arctic) using Roche 454 pyrosequencing, and they reported a total of 370 MOTUs, of which 294 belonged to Ascomycota, 54 to Basidiomycota, 2 to Zygomycota, and 20 to unknown fungi. Among these, Leotiomycetes, Dothideomycetes, and Eurotiomycetes were the major classes, with Helotiales, Capnodiales, and Chaetothyriales as the dominant orders, respectively. So far, only the study of Wang et al. [45] has presented metabarcoding data of the lichen-associated fungi, specifically focused on a single lichen species that had been collected across its distributional range, that is, the *Hypogymnia hypotrypa* in China. The authors identified 50 MOTUs comprising 28 Ascomycota, 11 Basidiomycota, 4 Zygomycota, 3 Chytridiomycota, and 4 unknown fungi. Fernández-Mendoza et al. [35] focused on the lichen species of alpine habitats and recovered abundant MOTUs of Dothideomycetes and Eurotiomycetes, and also found 25 MOTUs of Tremellomycetes (Basidiomycota). That study also showed that lichenicolous fungi could occur symptomless in the thalli of their known hosts, and could also be present in other lichen species. However, a large part of the MOTUs resulted as ‘unknown/uncultured ascomycetes/fungus’.

As these studies report a significant fraction of unknown fungi or uncultured fungi, further investigations and the improvement of reference sequence databases for more precise identification of fungi are still necessary. These MOTUs, indeed, could represent further and interesting extremotolerant species, which might be studied for their biological roles. It still remains questionable whether these taxa may represent obligatory lichen-associated fungi, as claimed by Peršoh and Rambold [46], or whether they are environmental taxa, the DNA of which have not been sequenced yet. The advantage of high throughput sequencing is the high diversity of sequence reads (usually over 10 times more MOTUs than with the culturing approaches). Despite the limited coverage of culture-based methods, these offer the possibility to isolate and study the fungal strains for their biological features. The culture-based and culture-free methods largely complement each other and are therefore both to be considered in studying the diversity of lichen associated fungi. Additionally, only direct microscopic observation can demonstrate whether the fungus grows and/or reproduces in the lichen (see below).

## 3. Systematics and Evolution of Extremotolerant Endolichenic Fungi

The observation of the diverse fungi with melanized cell walls in lichens had motivated research about their systematic relations. There were no doubts that melanized fungi did not form a single monophyletic lineage. However, the lack of teleomorphic states and the phenotypic variation of the mycelial characters hampered studies of systematics and diversity of these fungi. Taxonomic works classified numerous lichenicolous anamorphic species [47,48]. For many of them, no teleomorphic state was known and just the sequencing of the fungal isolates revealed new insights in the systematics of these lichen-associated fungi [49,50,51,52].

Multiple parallel lineages in the classes Dothideomycetes and Eurotiomycetes gave rise to more specialized animal and plant pathogens [53] (Figure 2). Their ancestors may have evolved in rock habitats during the dry climates of the late Devonian and middle Triassic, respectively [54]. Dothideomycetes is one of the most species rich classes in the Ascomycota, comprising about 19,000 species [55,56]. Within Dothideomycetes, a wide diversity of fungal life styles evolved, yet the order Capnodiales (in particular the family Teratosphaeriaceae) held the highest number of extremotolerant taxa (Figure 2). They were represented by the isolates from the rocks of the McMurdo Dry Valleys, Antarctica [57,58,59], high-altitudes of the Alps [60], hot deserts [61], and from grounds of salterns [62]. Interestingly, in this class, the two lichenized, monospecific genera *Cystocoleus* and *Racodium* were also found [63]. Within the Eurotiomycetes, the order Chaetothyriales comprises ecologically diverse extremotolerant fungi (Figure 2). Many of the fungi were feared for their pathogenic potential on the vertebrate hosts, including humans, where they could cause nasty chromoblastomycoses and phaeohyphomycoses. Yet, there were also aquatic, rock-inhabiting, ant-associated, and mycoparasitic, endophytic, and epiphytic life-styles, as well as species that could tolerate toxic compounds, which suggested a high degree of versatile extremotolerance [64]. The family Herpotrichiellaceae harbored a vast diversity of asexual morphs of the plant saprobic and clinically important species in cold- and warm-blooded vertebrates [64]. The main asexual genera were *Cladophialophora*, *Exophiala*, *Fonsecaea*, *Phialophora*, and *Rhinocladiella*. Interestingly, *Cladophialophora* and *Rhinocladiella* were also isolated from the lichens [27]. Other Chaetothyriales that were isolated from lichens formed their own monophyletic clades, which are awaiting a formal description [15,65].

## 4. Lichen Mycobiota in Axenic Culture

### 4.1. Isolation, Growth, and Diversity of Lichen Mycobiota in Culture

While the cultivation of the primary fungal symbiont was achieved soon after the discovery of the symbiotic nature of lichens in the second half of the 19th century, the presence of other fungi occurring asymptomatically in lichen thalli only received attention much later. The reason for the delayed research on the culurable mycobiome was certainly the erroneous interpretation of these fungi as contaminations of the culturing approach. Early studies of the asymptomatic lichen-associated mycobiota were provided by Petrini et al. [70] and Girlanda et al. [71], while in parallel, Crittenden et al. [72] isolated fungi from lichenicolous infections. Since the chemical surface sterilization was not efficient for Petrini et al. [70] because of the loose texture of the lichen tissues, the fragments of the shrubby lichens (*Cladonia* and *Stereocaulon* species) were only washed using sterile tap water. It could therefore not be guaranteed whether the fungi that were isolated from the lichens were residing inside the thalli or whether they just represented surface-attached spores [70]. Girlanda et al. [71] treated the lichen thalli with both sterile water and hydrogen peroxide prior to isolation from two lichen species (*Parmelia taractica* and *Peltigera praetextata*). Later, the reagents Tween20 and Tween80 were introduced in the isolation protocol of the lichen mycobionts and photobionts so as to wash away the bacterial and fungal contaminants from the thallus surface [73]. This step increased the success of isolating the primary mycobiont plus the potential additional asymptomatic fungal associates. According to Suryanarayanan et al. [74], who evaluated four procedures of surface sterilization of lichen thalli (washing with water, H_2_O_2_, ethanol, and NaOCl), the most reliable surface sterilization procedure was washing in 70% ethanol. When applying the sterilization procedures using only water and H_2_O_2_, the authors recorded a high rate of *Aspergillus* and *Penicillium* species, which they considered as epithalline, unspecific, and ubiquitous contaminants. When the surface sterilization was followed by washing the steps with ethanol and NaOCl, a broad spectrum of fungi were isolated, which indicated that the method eliminated the surface borne fungi (such as *Aspergillus* and *Penicillium*) and facilitated the growth of the asymptomatic, intrathalline, culturable fungi [65]. In addition, Arnold et al. [11] applied a gradient of washings in water, ethanol, and NaOCl, in order to specifically recover the intrathalline asymptomatic fungi from lichens. The efficiency of a surface sterilization procedure could be tested by pressing the surface-sterilized thallus fragments on a solid growth medium and monitoring any fungal growth after removal of the fragments. This important methodology for reducing contaminants has been so far rarely considered in the preparation of samples for metabarcoding analyses. The distinction of the epithalline and intrathalline fungi is highly debatable. In contrast to higher plants, lichens lack a cuticula as a protective layer. To take water up from their surfaces, lichens are necessarily open systems, and for the same reason, most lichen-inhabiting fungi are not confined to internal or external growth. An interesting case has been represented by *Cyphobasidium*, which seemed to be confined to the fungal upper cortex layer of the lichen [16].

The success of isolating fungi from the lichen thalli could be steered by the use of the different growth media on which the first inocula were placed (Figure 3). It was known that the different media compositions modulate the growth and the synthesis of secondary metabolites in the cultured lichen mycobionts [75]. Alternatively, only recently had the successful growth of diverse lichen-associated fungi been compared for different medium compositions and it was evaluated if the grown isolates, which represented different fungal classes, could it be specifically retrieved using certain media [65,76]. In this context, Muggia et al. [65] used six different growth media to expand the range of the culturable asymptomatic lichenicolous fungi from crustose, epilithic lichens and succeeded by isolating about 500 strains representing the four classes, namely, Eurotiomycetes, Dothideomycetes, Leotiomycetes, and Sordariomycetes. The media differed in the presence of the organic and inorganic compounds and were differently enriched by nutrients, such as sugars, metal compounds, amino acids, and vitamins [65]. The fungal growth was not dependent from the pH of the medium, but rather it was correlated with the presence of certain components in the media, which, indeed, more or less favored the development of certain fungal classes. Media containing magnesium, potassium, iron, and glucose were pivotal for the successful isolation of all of the fungal classes, whereas chloramphenicol, EDTA, and potassium hydroxide (KOH) reduced the successful isolation of Eurotiomycetes. The presence of asparagine, CaCl_2_, sodium, and peptone seemed to be important for the growth of Sordariomycetes. In addition, the abundant presence of metal ions in the medium seemed to favor the growth of the melanized strains belonging to the classes Eurotiomycetes and Dothideomycetes, which were easily cultivable on a malt yeast medium as well [46,65].

The eurotiomycetous and dothideomycetous, melanized strains from the lichens in many cases either corresponded or were phylogenetically closely related to the extremotolerant rock-inhabiting fungi [15,27,77]. However, several culture isolates from the different lichen hosts were identified as new lineages within the Eurotiomycetes [15,65] and seemed to be so far unique for lichen symbioses; these lineages are awaiting a formal species description. These undescribed taxa seemed to be preferentially distributed in the crustose thalli on rocks, but seemed to have a rather low specificity for their host species and geographic origins.

In contrast to the symptomless fungi, the symptom-causing lichenicolous fungi were very recalcitrant to grow axenically in culture and they needed to be isolated from spores, produced either in the sexual or asexual structures that were built on the host thallus. An isolation success was rather a rare event and it was usually reported after several attempts and a very long incubation time (sometimes in the range of years; L. *Muggia*, pers. comm.). Reliably identified isolates of symptomatic lichenicolous fungi were only available in culture collections since a few years ago [49,50,51,52,78,79]. In many cases, the spore germination occurred within a few days, but soon the mycelium ceased to grow. Whether the difficulty in isolating and culturing the symptomatic lichenicolous fungi was derived from the specificity and nutritional requirements that they showed towards their host lichens has still been largely untested. However, a relatively fast growth rate in culture of the symptomatic lichenicolous fungus *Capronia peltigerae* was correlated with its saprobic life style on the lichen host, and it was suggested that the other members of the genus would be easily cultured as well, if the recently collected and well dried material would be selected [78]. The cultures of the symptomatic lichenicolous fungi were the key for advanced systematic classification beyond the ITS phylogenies and for studying the biology and the metabolism of these extremotolerant taxa, which could also include strains producing interesting new secondary metabolites.

### 4.2. Lichen Mycobiota as Sources of New, Bioactive Secondary Metabolites

During the past decade, the fungi that were isolated from the lichen thalli were subjected to secondary metabolite analyses. Kellogg & Raja [83] have recently provided a detailed review on the progress done in the past decades in the analysis of the secondary metabolites that were produced by the asymptomatic lichenicolous fungi, and further information is available from yet another review [84]. So far however, only a very small fraction of the lichen-associated fungi were characterized for their secondary metabolite production. All of the characterized fungal strains were isolated from macrolichens, which formed fruticose or foliose thalli. These lichenicolous taxa, for which, so far, about 140 new chemical products were identified, were representative of a very widespread genera, such as *Aspergillus*, *Chaetomium*, *Penicillium*, *Sporomiella*, and *Trichoderma* [83]. Many other strains were characterized either only up to the genus level (e.g., *Coniochaete* sp., [85]; *Chrysosporium* sp., [86]) or not at all yet, and were only assigned a working number (e.g., the strain LL-RB0668, from which the bioactive lichenicolins A and B were isolated [86,87]). This spectrum of fungi indicated that generalists (either spores or mycelia), which grew fast in cultures, were mainly considered in secondary metabolite assays and that many slow-growing fungi, which were highly adapted to the lichen environment, had not been studied yet. This might have also explained why many specific lichenicolous basidiomycetes were not cultured yet, in particular *Cyphobasidiales* [16,35,88]. The common *Tremellales* with *Fellomyces* anamophs seemed to be isolated more easily from the lichens [89]. In some publications, the lichen hosts were not specified, which made the reproducibility of the results difficult, if not impossible. On the other hand, some studies succeeded in identifying the bioactivities of certain fungal strains but failed to relate them to a specific secondary metabolite [90].

In the pioneering works, thin-layer chromatography (TLC) was applied to directly detect the compounds that possibly originated from the symptomatic lichenicolous fungi [91] and shed first light into the chemical patterns that were involved in the fungal interaction. However, the TLC analysis offered only a qualitative overview of those compounds that were present in substantial amounts, whereas the bioactive molecules that were produced in lower concentrations may have remained undetected. Nowadays, more powerful tools of chromatography and spectroscopy, such as isotope labeling, mass spectrometry (MS), and MS/MS fragmentation patterns, as well as metabolomics and small-molecule protein interactomics analyses, have enhanced and facilitated the analyses of fungal compounds [90]. Thus, the described diversity of the secondary compounds that were produced by the asymptomatic lichenicolous fungal strains spanned from alkaloids, quinones, to aromatic compounds and peptides ([83] and the references therein). Many of these compounds showed antioxidant, cytotoxic, antibacterial, and antifungal properties and could be exploited for their pharmaceutical purposes against animal (including human) and plant pathogens [86,92,93]. However, none of the identified secondary metabolites had been used to develop therapeutics yet [83]. Beside their novelty, many of these secondary compounds were end-products of alternative, polyketide biosynthetic pathways and therefore presented diverse chemical structures [85,90].

## 5. Lichen-Associated Fungi and Their Interaction with Photobionts

Arnold et al. [11] noticed that the culturable, asymptomatic lichenicolous fungi were rarely isolated from the mycobiont layers of the lichen thallus (i.e., medulla, cortices), but were more often isolated from the photobiont layer. The co-culture experiments with the available fungal strains that were isolated from the lichens could also have been used to test the potential to interact in vitro with the photobionts of the host. Owing to an increasing interest in technological applications, co-cultures were recently reviewed and new strategies were proposed [94,95]. These approaches frequently also combined organisms that did not naturally form symbioses (e.g., [96]). The co-culture experiments with the lichen-associated black fungi had been performed previously, because these were the first and the most commonly isolated strains from the lichen thalli of the diverse geographic regions [15,27,28]. As their closest relative on the bare rock surfaces often formed subaerial biofilms with green algae and cyanobacteria, we suggested that they might have had an inherent affinity to algae. In their initial co-culture experiments, Gorbushina et al. [97] succeeded in demonstrating funga–algal interactions between several lichen photobionts species and rock-inhabiting fungi. TEM observations confirmed the close wall-to-wall contacts and the mucilage production around the contact zone between all of the tested rock-inhabiting fungi and the selected photobionts, while the formation of the haustorium-like structures was initiated only in very few fungal–algal combinations. Brunauer et al. [98] showed that an asymptomatic, chaetothyrialean, lichenicolous fungus interacted in subtle ways with the algae of the host lichens, which developed lichenoid structures by forming a continuous hyphal layer over the photobiont colonies and contacting the algal cells with appressoria [98]. These pioneering works inspired Ametrano et al. [99] to establish a series of co-culture experiments using species of the black fungal genus *Lichenothelia*. *Lichenothelia* was chosen because, since the time of its description [100,101], it was suggested to be a possible link between the lichenized and the non-lichenized life styles. Since the genus included both rock-inhabiting fungi, which were occasionally found growing with algae, and symptomatic lichenicolous fungi, it was critical to know more about the different lifestyles of the *Lichenothelia* species. Ametrano et al. [99] studied whether the growth of the *Lichenothelia* fungi would be enhanced by the photobiont presence when no organic carbon sources were provided in the medium, and whether the fungal–algal co-growth would have stimulated the formation of the lichen-like structures. The fungal growth rates were statistically evaluated and the structures of the mixed cultures were analyzed by the light and scanning electron microscopy. So far, the results showed that, in the *Lichenothelia*-photobiont system, the presence of the algae did neither influence the growth rate of the fungi nor the formation of any lichen-like structure [99].

Muggia et al. [95] developed a mixed culture system to combine the symptomatic lichenicolous fungus *Muellerella atricola* (isolated from the host lichen *Tephromela atra*) with two strains of the lichen photobiont genus *Trebouxia*. They noticed that the algal cells and fungal hyphae were arranged in a compact, layer-like structure and that the algal clumps were tightly bound by the hyphae. Nonetheless, no haustoria- or appressoria-type contacts between the hyphae and algae could be observed. Interestingly, the fungus produced asexual spores (conidia) in co-cultures was always adjacent and above the algal colonies, which was interpreted as the result of interacting signaling processes. This work represented a case of ‘enforced symbiosis’, where microorganisms that were not naturally occurring in this manner, needed to interact. Although the resulting phenotypes of enforced symbioses might not have been predicted, they represented interesting strategies for the extended screening programs in the search of novel bioactive compounds.

## 6. Conclusions

The number of studies focusing on lichen-associated fungi has increased rapidly in the past decades and has helped to uncover their diversity and potentials toward extremotolerance and interactions with algae. Both the culture-based and culture-independent methods have been developed and ad hoc improved in order to explain the chemical and genetic variation, as well as some of the evolutionary processes, which has led to the diversification of the symptomatic and asymptomatic lichenicolous fungi. There is still a lot work ahead in order to understand the specificity and biology of these fungi, as the number of strains that have been isolated in culture and fully characterized is still low. Preliminary experiments have showed that numerous fungi have a potential to build sustained interactions with algae or to modulate the chemical traits of the lichen holobiome. It will therefore be of general interest to characterize the metabolome and the interactome of these taxa. Furthermore, the lichen-associated fungi may be seen as unexplored hotspots for the discovery of new chemical compounds. Perspectives for exploring the world of the lichen-associated fungi more deeply are promising because of the advancements in high throughput sequencing techniques and in the setups for co-culture experiments. As the extremotolerant members of the lichen mycobiome seem to cope well with the poikilohydric conditions of their lichen hosts, it will be important to study their activities in the symbiotic holobiont under different hydration regimes.

## Figures and Tables

**Figure 1 life-08-00015-f001:**
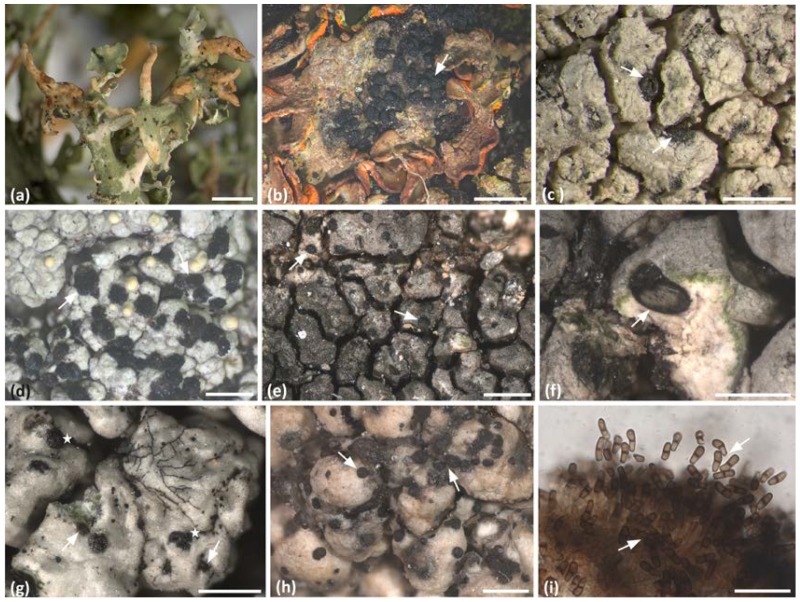
Habit of lichenicolous fungi on their lichen host. (**a**) *Tremella* sp. on *Cladonia furcata*; (**b**) A, *Rhagadostoma lichenicola* on *Solorina crocea*; (**c**) *Sagediopsis fissurisedens* on *Aspilidea myrinii*; (**d**) *Sclerococcum sphaerale* on *Pertusaria corallina*; (**e**) *Endococcus perpusillus* on *Schaereria fuscocinerea*; (**f**) *Rosellinula haplospora* on *Aspicilia caesiocinerea*; (**g**) *Lichenodiplis lecanorae* on *Tephromela atra* (stars labelling the pycnidia of the lichen mycobiont); (**h**) *Minutoexcipula tuerkii* on *Pertusaria glomerata*; and (**i**) detail of sporodochium and conidia (arrow) of *Minutoexcipula tuerkii* on *Pertusaria glomerata*. Arrows point to the perithecia (**b**,**c**,**e**,**f**,) and sporodochia (**d**,**g**,**h**) of the lichenicolous fungi. Scale bars: (**a**,**b**) = 2 mm, (**c**) = 1 mm, (**d**–**h**) = 0.5 mm, and (**i**) = 20 μm.

**Figure 2 life-08-00015-f002:**
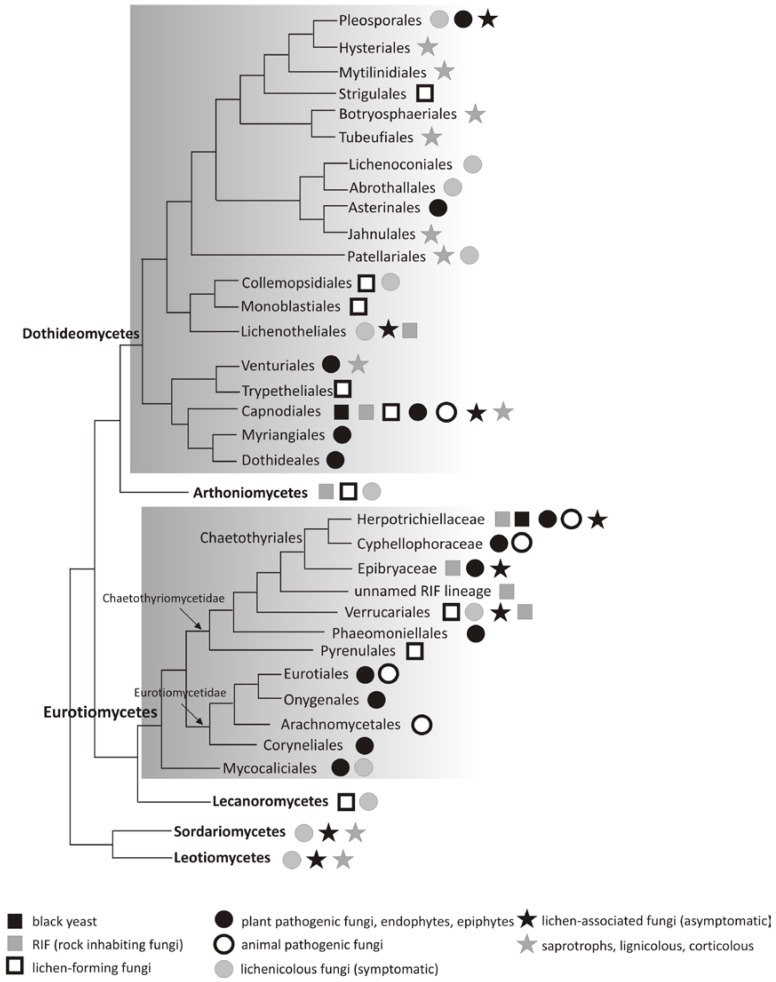
Schematic phylogenetic representation of major lineages in which lichen-associated fungi are found. The phylogeny was graphically reconstructed merging information from most recent phylogenetic studies [64,66,67,68,69].

**Figure 3 life-08-00015-f003:**
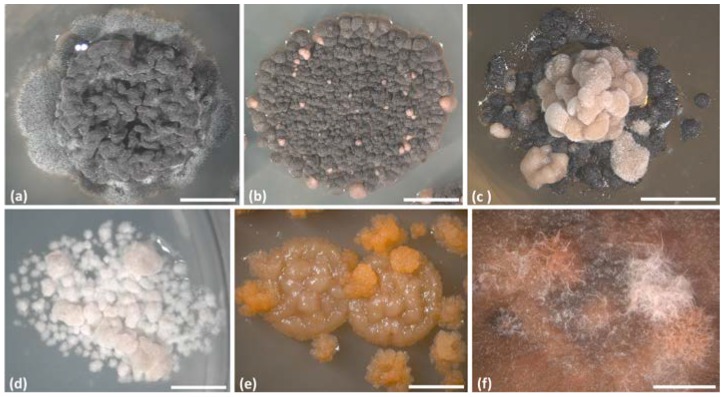
Habit of axenically isolated lichen-associated fungi. The sample ID, its phylogenetic placement (class and/or order or lineage, sensu Muggia et al. [15,65]) and the acronym of the medium on which it grows are reported. (**a**) A1085, Leotiomycetes, DG18; (**b**) A1148, Eurotiomycetes, Chaetothyriales, clade VI, SAB; (**c**) A1073, Dothideomycetes, Myriangiales, DG18; (**d**) A1153, Eurotiomycetes, *Sclerococcum*-clade, LBM; (**e**) A1109, Eurotiomycetes, Chaetothyriales, clade VI, MY; and (**f**) A1074, Dothideomycetes, Pleosporales, DG18. The different colours of the mycelia in B and C are derived from a variation in melanization and belong to the same fungus. Growth medium acronyms: DG18, Dichloran/Glycerol agar [80]; LBM, Lilly & Barnett medium [81]; MY, Malt Yeast-extract [81]; SAB, Sabouraud [82]. Scale bars: (**a**–**d**,**f**) = 4 mm, and (**e**) = 2 mm.

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
