# Peer review of "Fungal Diversity in Lichens: From Extremotolerance to Interactions with Algae"

_life, 2018, doi:10.3390/life8020015_

Round 1

Reviewer 1 Report

The present manuscript is a review on endoliochenic fungi, including both methodological aspects for their detection and considerations about their phylogenetic relationships, ecological roles, and potential applications. The review is written by two experts in the field and is overall sound and detailed. I have annotated a word version (attached) and here repeat my main concerns with the paper:

The title suggests a focus on extremotolerant endolichenic fungi and their potential interactions with photobionts, but the paper is a general review on the topic of endolichenic fungi, with no clear focus on what is stated in the title. In fact, extremotolerant fungi are only mentioned and focused on in a few places. This could be remedied in two ways: either change the title (rather straightforward) or adjust the structure and content of the review, which is more complicated, as it will require to shorten the general aspects and expand specifically on extremotolerant fungi.The problem is particularly noticeable when going from chapter 2 (which fits the title) to the remaining chapters which are then again mostly concerned with general issues regarding all endolichenic fungi. Also the conclusions are those of a general review on the topic and do not at all refer to the more specific aspects of the title.

A clear definition of what extremotolerant means in this context is missing, which is part of the structural problem of the text. In part, extremotolerant is equalled to the environment of the host lichens, in part it is equalled with fungi that have pigmented (melanized) walls. The latter certainly cannot be taken as a equivalent of extremotolerant, and simply rock-inhabiting is also not necessarily an extrem. So if the focus is on these fungi, either a clear definition of extremotolerant is required, based on pertinent literature, or else it should be replaced by another definition, such as "black fungi", which is apparently the morphodeme the authors intend to focus on. However, the latter might not make much sense from a practical point of view, since pigmented hyphae evolved multiply within the fungi and these fungi neither form phylogenetic nor ecological entities. So my suggestion is to broaden the title and maybe focus in one chapter at the end on the melanized fungi.

While the text shows broad insight into the topic, the authors missed some important, specific details and also some important references. I have annotated these instances in the text. Also, in parts the text is descriptive rather than analytical and rewording could make it both more concise and more specific to a particular topic.

The English is overall OK but has occasional problems in style and grammar. I have annotated many of these but perhaps a check by a native speaker is required.

Author Response

Dear Editor, dear Reviewers,

please find here below in bold a point-by-point response to your comments.

Reviewer 1. The present manuscript is a review on endoliochenic fungi, including both methodological aspects for their detection and considerations about their phylogenetic relationships, ecological roles, and potential applications. The review is written by two experts in the field and is overall sound and detailed. I have annotated a word version (attached) and here repeat my main concerns with the paper:

DONE. We considered all the annotations in the text. Among these, we accepted to move the original chapter "3. Diversity as revealed by high-throughput sequencing" ahead, and it now arranged as chapter 2.

The title suggests a focus on extremotolerant endolichenic fungi and their potential interactions with photobionts, but the paper is a general review on the topic of endolichenic fungi, with no clear focus on what is stated in the title. In fact, extremotolerant fungi are only mentioned and focused on in a few places. This could be remedied in two ways: either change the title (rather straightforward) or adjust the structure and content of the review, which is more complicated, as it will require to shorten the general aspects and expand specifically on extremotolerant fungi. The problem is particularly noticeable when going from chapter 2 (which fits the title) to the remaining chapters which are then again mostly concerned with general issues regarding all endolichenic fungi. Also the conclusions are those of a general review on the topic and do not at all refer to the more specific aspects of the title.

DONE. The title was changed, trying to make it more general as follow: Fungal diversity in lichens: from extremotolerance to interactions with algae"

A clear definition of what extremotolerant means in this context is missing, which is part of the structural problem of the text. In part, extremotolerant is equalled to the environment of the host lichens, in part it is equalled with fungi that have pigmented (melanized) walls. The latter certainly cannot be taken as a equivalent of extremotolerant, and simply rock-inhabiting is also not necessarily an extrem. So if the focus is on these fungi, either a clear definition of extremotolerant is required, based on pertinent literature, or else it should be replaced by another definition, such as "black fungi", which is apparently the morphodeme the authors intend to focus on. However, the latter might not make much sense from a practical point of view, since pigmented hyphae evolved multiply within the fungi and these fungi neither form phylogenetic nor ecological entities. So my suggestion is to broaden the title and maybe focus in one chapter at the end on the melanized fungi.

DONE. We have now modified a bit the Introduction explaining the meaning of "extremotolerance". We refrain though to add a paragraph specifically focusing on black fungi, as the different aspects of their diversity is recurrently treated in the manuscript in each part dealing with phylogeny, fungal culture and the interactions with algae.

While the text shows broad insight into the topic, the authors missed some important, specific details and also some important references. I have annotated these instances in the text. Also, in parts the text is descriptive rather than analytical and rewording could make it both more concise and more specific to a particular topic.

DONE. We have revised the text and reworded where it was suggested and in few more parts also according to the suggestions of Reviewer 2. Some citations have been also added and therefore the numeration of the References has been congruently updated.

The English is overall OK but has occasional problems in style and grammar. I have annotated many of these but perhaps a check by a native speaker is required.

DONE

Reviewer 2 Report

A review about the present stage of study about fungi in extremotolerant lichens is present. It well written and illustrated, and can be smoothly taken up by the reader. It clearly shows the benefits and limitations of metabarcoding as well as the increased need of improved traditional cultivation approaches.

Generally a comma is to be used before and after “therefore”. In most cases “build” should better be replaced with “form”.

Minor points are suggested for improvement for the following lines:

23 delete “human opportunists”, since this is not an ecological niche, but a blind end without chance of the fungus to spread further

Figure 2: Write Hysteriales instead of Histeriales, Trypetheliales instead of Trypeteliales

Page 6: The insufficient resolution of the whole ITS and the insufficient surface disinfections in the metabarcoding approaches should be mentioned among the shortcomings.

150 delete “fungi”

151 “classes Sordariomycetes”, not “classed of Sordariomycetes”

169 “blasted”: can the abbreviation BLAST (basic local alignment tool) be used as verb? I recommend not using lab slang in scientific writing.

176 insert “or even contaminations” behind “environmental taxa”

176 taxa, which have not been sequenced: only macromolecules can be “sequenced” but not a taxon. Here again lab slang should be avoided.

178 insert “in” before “culture approaches”

178 delete “do)”

181 change “themselves” to “each other”

182 It is recommended concluding the paragraph about the limitations of metabarcoding with a sentence like this: “Only direct microscopic observation can demonstrate whether the fungus grows and/or sporulates in the lichen”

186 insert “century” after “the 19th”

213 The authors may insert a sentence like this” This important methodology for reducing contaminants is rarely considered in metabarcoding”, or emphasize this in the paragraph about metabarcoding more clearly

225 representing

228 change “dependent by the pH” to “dependent from the pH”

228 delete “it” before “ rather”

240 change “samples ID” to “sample ID”

267 members

269 Since with cultures additional DNA regions can be used for phylogenies and species delimitations, something could be inserted behind “studying the”, e.g. a phrase like “advanced systematic classification beyond ITS phylogenies,”

285 lichenolins

287 slowly growing or slow-growing

288 replace “contemplated” with “considered”

290 Many Tremella species are highly specific to their host lichens. Better change “ubiquitous” to “common” before “Tremellales”

290 seem

290 seem to be isolated more easily

304 alkaloids

366 change “used” to “unexplored”

In the conclusions section, it would be good to pick up the term “extremotolerant” from the title again.

Author Response

Dear Editor, dear Reviewers,

please find here below in bold a point-by-point response to your comments.

Reviewer 2.

A review about the present stage of study about fungi in extremotolerant lichens is present. It well written and illustrated, and can be smoothly taken up by the reader. It clearly shows the benefits and limitations of metabarcoding as well as the increased need of improved traditional cultivation approaches.

Generally a comma is to be used before and after “therefore”. In most cases “build” should better be replaced with “form”. Minor points are suggested for improvement for the following lines:

23 delete “human opportunists”, since this is not an ecological niche, but a blind end without chance of the fungus to spread further. DONE.

Figure 2: Write Hysteriales instead of Histeriales, Trypetheliales instead of Trypeteliales. DONE. Also, the family name Phaemoniellaceae has been changed into the order name Phaeomoniellales.

Page 6: The insufficient resolution of the whole ITS and the insufficient surface disinfections in the metabarcoding approaches should be mentioned among the shortcomings. DONE.

150 delete “fungi” DONE.

151 “classes Sordariomycetes”, not “classed of Sordariomycetes” DONE.

169 “blasted”: can the abbreviation BLAST (basic local alignment tool) be used as verb? I recommend not using lab slang in scientific writing. DONE. We changed into "resulted as".

176 insert “or even contaminations” behind “environmental taxa” DONE.

176 taxa, which have not been sequenced: only macromolecules can be “sequenced” but not a taxon. Here again lab slang should be avoided. DONE. We changed into " the DNA of which have not been sequenced yet".

178 insert “in” before “culture approaches”. DONE, changed according the Reviewer 1's suggestion "than with culturing approaches".

178 delete “do)” DONE.

181 change “themselves” to “each other” DONE.

182 It is recommended concluding the paragraph about the limitations of metabarcoding with a sentence like this: “Only direct microscopic observation can demonstrate whether the fungus grows and/or sporulates in the lichen” DONE.

186 insert “century” after “the 19th” DONE.

213 The authors may insert a sentence like this” This important methodology for reducing contaminants is rarely considered in metabarcoding”, or emphasize this in the paragraph about metabarcoding more clearly. DONE.

225 representing DONE.

228 change “dependent by the pH” to “dependent from the pH” DONE.

228 delete “it” before “ rather” DONE.

240 change “samples ID” to “sample ID” DONE.

267 members DONE.

269 Since with cultures additional DNA regions can be used for phylogenies and species delimitations, something could be inserted behind “studying the”, e.g. a phrase like “advanced systematic classification beyond ITS phylogenies,” DONE.

285 lichenolins DONE.

287 slowly growing or slow-growing DONE, changed into slow-growing.

288 replace “contemplated” with “considered” DONE, changed into "studied".

290 Many Tremella species are highly specific to their host lichens. Better change “ubiquitous” to “common” before “Tremellales” DONE.

290 seem DONE.

290 seem to be isolated more easily DONE.

304 alkaloids DONE.

366 change “used” to “unexplored” DONE.

In the conclusions section, it would be good to pick up the term “extremotolerant” from the title again. DONE, the first sentence was changed as follow: "The number of studies focusing on lichen-associated fungi has increased rapidly in the past decades and helped to uncover their diversity and potentials toward extremotolerance and interactions with algae."

Round 2

Reviewer 1 Report

The authors have done an excellent job addressing previous reviewers comments and I am now very pleased with this manuscript, which is an important contribution to the field. I have annotated still some minor points in the attached manuscript file that should be easy to fix.

Author Response

We thank the Reviewer!

we have addressed all the suggested changes and resubmit a clean version of the manuscritp.
